



# The roles of the Quasi-Biennial Oscillation and El Nino for entry stratospheric water vapour in observations and coupled chemistry-ocean CCMI and CMIP6 models

Shlomi Ziskin Ziv [1,2], Chaim I. Garfinkel [3], Sean Davis [4], and Antara Banerjee [4,5]

[1]Department of Physics, Ariel University, Ariel, Israel.
[2]Eastern R&D center, Ariel, Israel
[3]The Fredy and Nadine Herrmann Institute of Earth Sciences, Hebrew University of Jerusalem, Jerusalem, Israel
[4]NOAA Chemical Sciences Laboratory, Boulder, CO, USA
[5]Cooperative Institute for Research in Environmental Sciences.

**Correspondence:** Shlomi Ziskin Ziv (shlomiziskin@gmail.com)

**Abstract.** The relative importance of two processes that help control the concentrations of stratospheric water vapor, the Quasi-Biennial Oscillation (QBO) and El Niño-Southern Oscillation (ENSO), are evaluated in observations and in comprehensive coupled ocean-atmosphere-chemistry models. The possibility of nonlinear interactions between these two is evaluated both using Multiple Linear Regression(MLR) and three additional advanced machine learning techniques. The QBO is found

to be more important than ENSO, however nonlinear interactions are non-negligible, and even when ENSO, the QBO, and potential nonlinearities are included the fraction of entry water vapor variability explained is still substantially less than what is accounted for by cold point temperatures. While the advanced machine learning techniques perform better than an MLR in which nonlinearities are suppressed, adding nonlinear predictors to the MLR mostly closes the gap in performance with the advanced machine learning techniques. Comprehensive models suffer from too weak a connection between entry water and

the QBO, however a notable improvement is found relative to previous generations of comprehensive models. Models with a stronger QBO in the lower stratosphere systematically simulate a more realistic connection with entry water.

## 1 Introduction

Water vapor (WV) provides most of the greenhouse effect in the atmosphere, and of the total water vapor feedback to increasing anthropogenic greenhouse gas emmissions, roughly $10\%$ is associated with water vapor in the stratosphere (Forster and Shine,

1999; Solomon et al., 2010; Dessler et al., 2013; Wang et al., 2017; Banerjee et al., 2019). The amount of water vapor that enters the stratosphere is also important for stratospheric chemistry and specifically the severity of ozone depletion (Solomon et al., 1986). Hence, it is important to understand the factors that control the entry of water vapor into the stratosphere on all timescales, and to consider whether comprehensive models used for ozone and climate change assessments represent these factors correctly.

Most of the water vapor in the lower stratosphere transited from the tropical upper troposphere through the tropical tropopause, and therefore tropical temperatures near the cold point largely determine lower stratospheric water vapor concentrations (Mote





et al., 1996a; Zhou et al., 2004, 2001; Fueglistaler and Haynes, 2005b; Fueglistaler et al., 2009; Randel and Park, 2019). Many processes have been shown to modulate these cold point temperatures, and the goal of this work is to re-evaluate the influence of these processes, and in particular their nonlinear interactions, on entry water vapor. We then consider the ability of comprehensive models to represent this effect.

The QBO modulates water vapour mixing ratios in air entering the stratosphere through the its influence on temperatures in the tropical tropopause region (Reid and Gage, 1985; Zhou et al., 2001, 2004; Fujiwara et al., 2010; Liang et al., 2011; Kawatani et al., 2014). Specifically, warmer cold point temperatures during the QBO phase with westerlies near 50hPa (hereafter wQBO) lead to moistening, and colder temperatures during the QBO phase with easterlies near 50hPa (eQBO) lead to drying of the stratosphere. Comprehensive climate models typically struggle to capture the downward propagation of the QBO to the lower

stratosphere (Rao et al., 2020a; Richter et al., 2020), and consistent with this Smalley et al. (2017) found that the Chemistry Climate Model Validation Activity 2 (CCMVal2) models and most of the the Chemistry-Climate Model Initiative (CCMI, Morgenstern et al., 2017) models they considered struggle to capture an influence of the QBO on entry water.

El Niño (EN), the ENSO phase with anomalous warming of the tropical Eastern Pacific, has been shown to lead to a cooler tropical lower stratosphere and warmer tropical troposphere (Free and Seidel, 2009; Calvo et al., 2010; Simpson et al., 2011).

In addition, EN also forces a Rossby wave response that extends to the tropopause, whereby anomalously cold temperatures are present over the Central Pacific, and anomalously warm temperatures are present over the Indo-Pacific warm pool (Yulaeva and Wallace, 1994; Randel et al., 2000; Zhou et al., 2001; Scherllin-Pirscher et al., 2012; Domeisen et al., 2019). This zonal dipole in temperature has been shown to affect water vapor below the cold point: water vapor decreases in the region with cold anomalies and increases in the region with warm anomalies by $\sim 25\%$ (Gettelman et al., 2001; Hatsushika and Yamazaki,

2003; Konopka et al., 2016).

The net effect of these temperature anomalies on tropical lower stratospheric water vapor is complex. While the lower stratosphere clearly was moister following the two largest EN events in the satellite era (in 1997/1998 and in 2015/2016) (Fueglistaler and Haynes, 2005a; Avery et al., 2017), moistening also was evident following two of the strongest La Nina events (in 1998/1999 and 1999/2000). The impact of more moderate events is less clear, and any net effect is not statistically

significant considering the shortness of the data record (Garfinkel et al., 2018, 2021). There is no consensus among models as to the sign of the impact of ENSO on water vapor, with many models predicting a response opposite to that observed (Garfinkel et al., 2021).

Finally, the strength of Brewer Dobson circulation has been found to be important in determining entry water vapor, with a faster circulation associated with cooling of the cold point and dehydration (Randel et al., 2006; Dessler et al., 2013; Dessler

et al., 2014; Ye et al., 2018).

The net response to these various forcings is often quantified using multiple linear regression (e.g. Diallo et al., 2018; Tian et al., 2019), which implicitly assumes that the response to these forcings is linear, i.e. that the response to a given magnitude El Niño is equal and opposite to that of a La Niña event of equal magnitude. This technique also assumes that the response to e.g. ENSO and QBO is the sum of the linear responses to each individual forcing. Recent work has pointed out two faults of

such an assumption. First, Garfinkel et al. (2018) found that the response to ENSO is nonlinear, and hence such a methodology





may underestimate the impact of ENSO on stratospheric water vapor. Second, Yuan et al. (2014) find that the QBO has larger amplitude and longer period during La Niña conditions than during El Niño. Hence the difference between the warmer CPT temperatures during wQBO and colder temperatures during eQBO are larger during La Niña than during El Niño. This strengthens earlier findings that the greatest dehydration of air entering the stratosphere from the troposphere occurs during

the winter under La Niña and easterly QBO conditions (Zhou et al., 2004; Liang et al., 2011). Specifically, Yuan et al. (2014) argue that the net effect of ENSO and the QBO is not just a linear superposition of their independent influences, but the net result of their mutual interaction.

    The goal of this work is to reconsider the relative importance of the QBO and ENSO while taking into consideration the possibility for nonlinearity in the response, and to then consider whether the most-recent comprehensive models are capable of

simulating the response. After introducing the data and methodology in 2, we evaluate the relative sucesses of a Multiple Linear Regression(MLR) and of more advanced machine learning (ML) techniques with ENSO, QBO, BDC, and mid-tropospheric temperature as predictors, in an attempt to find the factors that most succinctly explain observed water vapor variability. We also consider the fraction of entry water vapor variability that can be accounted for by variations of the cold point temperature as an upper bound on how much water vapor variability is predictable from large scale processes. We then add two nonlinear

predictors to the MLR, and demonstrate that they are as important as e.g. a linear ENSO predictor. Finally, we consider the ability of comprehensive coupled ocean-atmosphere-chemistry models to simulate the connection between the QBO and entry water.

## 2   Data and methodology

### 2.1   Data

The **S**tratospheric **W**ater and **O**z**O**ne **S**atellite **H**omogenized data set (SWOOSH) (Davis et al., 2016) features a merged, gridded, homogenized and filled water vapor product from various limb sounding and solar occultation satellites over the previous ~30 years. The measurements are monthly means comprised of the following instruments: SAGE-II/III, UARS HALOE, UARS MLS, and Aura MLS. We use the zonal mean product (latitude, pressure) and the 3D (latitude, longitude, pressure) product as described in Table 1. The former has a high latitudinal resolution of $2.5°$ and extends to the HALOE period (1990s), while

the latter has a horizontal resolution of $20° \times 5°$ but relies on the high sampling rates available with AURA since 2004. While the latter data set does include some data as early as 1994, there are many gaps and filling these gaps in a self-consistent way is out of the scope of this analysis. Both data sets have a pressure level range of 300 to 1 hPa though our focus is on entry water at 82hPa and 68hPa. We use the zonal mean product when focusing on zonal mean entry water, and the 3D product when showing lat-lon maps of regression coefficients.

We examine six models participating in the CCMI and five coupled chemistry-climate models participating in sixth phase of the Coupled Model Intercomparison Project (CMIP6; Eyring et al., 2016). We only include CMIP6 models with interactive stratospheric chemistry as such a coupled chemistry-climate configuration has been shown to lead to more robust interannual variability of temperatures in the lower stratosphere as compared to models with fixed ozone (Yook et al., 2020). Note that



**Table 1.** Description of the target used in this analysis.

| Target field | SWOOSH field name | SWOOSH file name | years used |
|---|---|---|---|
| zonal mean | combinedanomfillanomh2oq | swoosh-v02.6-198401-201912-latpress-2.5deg-L31 | 1994-2019 |
| 3D | combinedanomh2oq | swoosh-v02.6-198401-201912-lonlatpress-20deg-5deg-L31 | 2005-2019 |

most of the models nevertheless simulate a too-warm cold point and too-little interannual variability of entry water (Garfinkel et al., 2021).

CCMI-1 was jointly launched by the International Global Atmospheric Chemistry (IGAC) and the Stratosphere-troposphere Processes And their Role in Climate (SPARC) to better understand chemistry-climate interactions in the recent past and future

climate (Eyring et al., 2013; Morgenstern et al., 2017). We analyze the Ref-C2 simulations, which span the period 1960-2100, impose ozone depleting substances as specified by the World Meteorological Organization (2011), and impose greenhouse gases other than ozone depleting substances as in Representative Concentration Pathway (RCP) 6.0 (Meinshausen et al., 2011). More details about these simulations are included in Eyring et al. (2013). We only consider CCMI models with a coupled ocean, and we include all available ensemble members. The CCMI-1 models used in this study are listed in Table 2. CCMI-2 models

are instructed to nudge the QBO rather than spontaneously simulate it. While this nudging should lead to an improved ability to capture the temperature response to the QBO (as discussed in section 4), this improvement is not because the models themselves are necessarily better and nudging is known to interfere with the transport of trace gases (Orbe et al., 2017, 2018). Hence the water vapor variability in CCMI-2 models is outside the scope of this study.

In addition to the CCMI-1 models, we also consider five Earth System models with coupled chemistry that are participating

in CMIP6: CESM2-WACCM (Gettelman et al., 2019), GFDL-ESM4 (Dunne et al., 2019), CNRM-ESM2-1 (Séférian et al., 2019), MRI-ESM2-0 (Yukimoto et al., 2019), and UKESM1-0-LL (Sellar et al., 2019). The climatology and seasonal cycle of stratospheric water vapor in these models is documented in Keeble et al. (2020). For all CMIP6 models we focus on the historical integrations of the period 1850 to 2014. Note that standard CMIP6 output includes the 70hPa and 100hPa levels but unfortunately no level in-between, and so our ability to diagnose physical processes near the cold point is limited. (In contrast,

CCMI output is available both near 80hPa and 90hPa.) All eleven models spontaneously represent the QBO (Rao et al., 2020a; Richter et al., 2020; Rao et al., 2020b) though as discussed in section 4 the quality of the simulation varies. In total, more than 2500 year of model output are available. All data are deseasonalized by subtracting the long term monthly means for that specific data product.

## 2.2 Target Variables and Indices

The target variable for all data sources is entry water, defined as water vapor at 82 hPa for SWOOSH, the closest archived level to 82hPa for CCMI, and 70hPa for CMIP6. The Quasi-Biennial Oscillation index is derived from the 50 mb zonal wind data in the NCEP/NCAR Reanalysis Climate Data Assimilation System (Climate Prediction Center, 2012). While including levels lower than 50hPa may lead to a slight improvement of the fit in observational data, many of the CCMI/CMIP6 models struggle



Table 1: Data products used

| | data source | ensemble members | reference |
|---|---|---|---|
| obs | SWOOSH v2.6 | 1 | Davis et al. (2016) |
| | ERA-5 | 1 | Hersbach et al. (2020) |
| CCMI | NIWA-UKCA | 5 | Morgenstern et al. (2009) |
| | CESM1 WACCM | 3 | Garcia et al. (2017) |
| | CESM1 CAM4-chem | 3 | Tilmes et al. (2016) |
| | HadGEM3-ES | 1 | Hardiman et al. (2017) |
| | MRI-ESM1r1 | 1 | Yukimoto et al. (2012) |
| | EMAC-L47MA | 1 | Jöckel et al. (2016) |
| CMIP6 | CESM2-WACCM | 1 | Gettelman et al. (2019) |
| | GFDL-ESM4 | 1 | Dunne et al. (2019) |
| | CNRM-ESM2-1 | 1 | Séférian et al. (2019) |
| | MRI-ESM2-0 | 1 | Yukimoto et al. (2019) |
| | UKESM1-0-LL | 1 | Sellar et al. (2019) |

**Table 2.** The data sources used in this study. For CMIP6 models we focus on the historical integrations of the period 1850 to 2014, and for CCMI the Ref-C2 simulations spanning the years 1960 to 2100.

to capture any remnant of the QBO below 50hPa (Rao et al., 2020a; Richter et al., 2020; Rao et al., 2020b) and hence we use 50hPa only throughout this paper. The lagged correlation of the QBO with near-82hPa entry water area averaged between 15S and 15N is shown in Figure 1a, and it is clear that a lag of 2 to 5 months maximizes the relationship in observations and in models. In Section 3 we use a lag of 5 months, and in Section 4 we use a lag of 2 months for CCMI and 5 months for CMIP6, though results are similar if the lag is changed by a few months. A later lag for the QBO is used for CMIP6 due to the difference in available pressure levels used to define entry water.

The El-Niño Southern Oscillation is tracked using the NINO3.4 index (5° North-5°South, 170°-120°West) sourced with ERSSTv5 data with a 1981-2010 base period. The data is taken from NOAA (Climate Prediction Center, 2012).

The CCMI and CMIP6 integrations include both long-term changes due to climate change in addition to the interannual variability which is our focus. The analyses in section 4 therefore includes, in addition to the QBO regressor, a regressor to track greenhouse gas concentrations (the equivalent $CO_2$ from the RCP6.0 scenario and historical $CO_2$ concentrations for historical simulations, Meinshausen et al., 2011). For the observational analysis, we do not include a $CO_2$ regressor, but instead





detrend all time-series, for two reasons: first, the regression coefficient for $CO_2$ in an MLR is extremely sensitive to whether we include the HALOE data or not, and, second, the ML methods are more stable when provided with fewer predictors on which to train the model. Both of these effects likely arise because of the short duration of the observational data-record.

The T500 index is the air temperature at 500 hPa averaged over the tropics (20°S to 20°N) taken from the ERA5.1 reanalysis
(Copernicus Climate Change Service, 2017; Hersbach et al., 2020). The BDC (Brewer–Dobson Circulation) index is the ERA5 variable "mean temperature tendency due to parametrisations" at 70 hPa averaged over the tropics (20°S to 20°N). In the deep tropics, the dominant contribution to the "mean temperature tendency due to parametrisations" is radiative heating.

The cold point temperature (CPT) index is calculated as in Randel and Park (2019), who use air temperature data from three equatorial radiosonde stations: Nairobi (1°S, 37°E), Manaus (3°S, 60°W), and Majuro (7°N, 171°E) sourced from the
Integrated Global Radiosonde Archive (IGRA) (Durre et al., 2006).

Note that the correlation of the BDC with the QBO is -0.66 (Figure 2), and hence including both in a single regression or ML model can lead to overfitting. If the BDC is defined at 82hPa (instead of at 70hPa) the correlation with the QBO drops, but then the correlation of the BDC with cold point temperatures reaches -0.72 over the period since 2005. Hence there is again the potential for overfitting if both are included, and if only the BDC is included there is ambiguity as to whether a signal is
due to the BDC or rather actually is associated with CPT but appears in the BDC regression coefficient because of the tight relationship between the CPT and BDC. Finally, the correlation between T500 and ENSO is 0.52, and if we high-pass filter the data to focus on interannual timescales the correlation increases further. Hence there is a similar risk of overfitting if both are included in a MLR, and similar ambiguity if only one is included.

All indices are deseasonalized by removing the long term monthly means. We do not consider seasonality in this work in
order to maximize the degrees of freedom, though we certainly acknowledge that the regression coefficients for, say, ENSO change sign between midwinter and late spring (Garfinkel et al., 2018, 2021). For all of these predictor time series, we divide by the standard deviation before constructing a MLR or ML model.

As discussed in Randel and Park (2019), cold point temperatures are highly correlated with entry water vapor (correlation of $\sim 0.8$ from 1993-2017 for 60S-60N averaged entry water vapor). This result is reproduced here over the period 2005-2019, but
showing the latitude vs. longitude distribution, in Figure 3a. We allow the CPT to lead entry water vapor by up to five months. Correlations peak above 0.8, and more generally 75% (i.e. the maximum $R^2$ on Figure 3a) of the cold point temperature and entry water variability are linearly related. We treat this 0.8 correlation as an upper bound on the effect that large scale, monthly mean dynamics can have on entry water vapor (with the remaining 25% due to processes on smaller spatial scales or shorter timescales). The aim of this paper is to understand the 75% of the variability that is due to large scale processes. In particular
to what extent can this 75% of the variability in turn be explained by large scale processes remote to the cold point such as the QBO and ENSO?

## 2.3 Machine Learning (ML) models

As discussed in the introduction, the connection between the QBO, ENSO, and entry water is not necessarily linear. Accordingly, we pick three popular types of ML models which we use in a supervised learning regression: Support Vector Machines





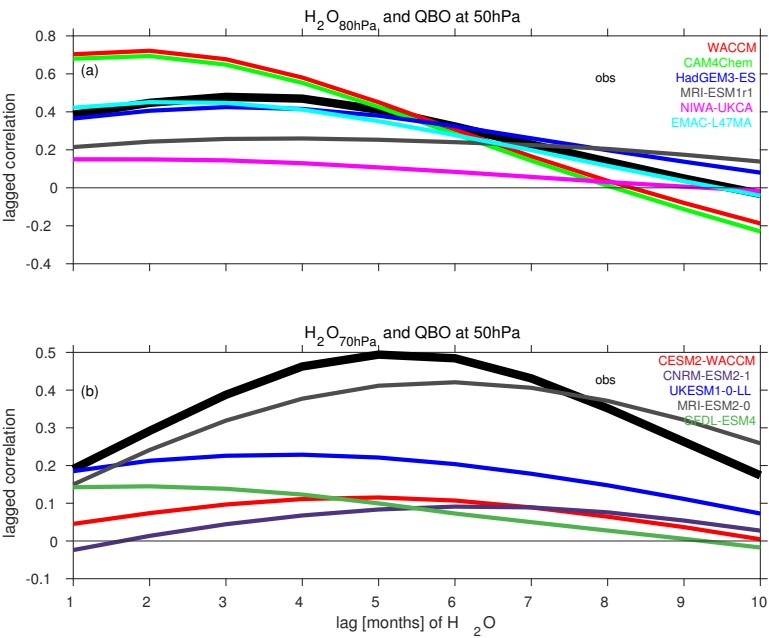

**Figure 1.** Lagged correlation between the QBO at 50hPa and tropical water vapor at (a) 80hPa in CCMI models and (b) 70hPa in CMIP6 models. The combinedanomfillanomh2oq product of swoosh-v02.6-198401-201912-latpress-2.5deg-L31 is used for observations from 1994 to 2019.

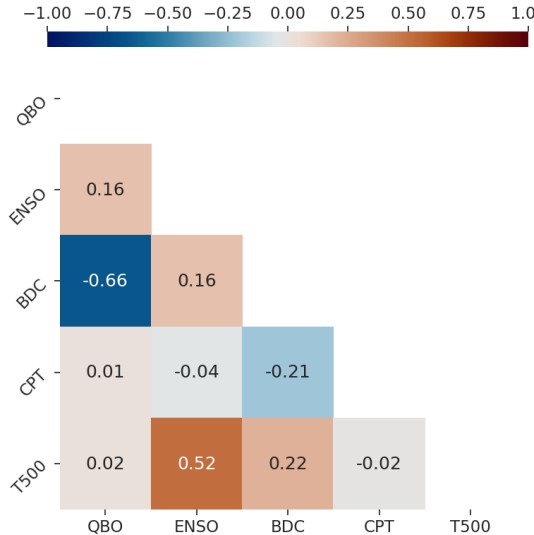

**Figure 2.** A correlation heatmap for the predictors used in the analysis. The time-span is from 1994 to 2019.



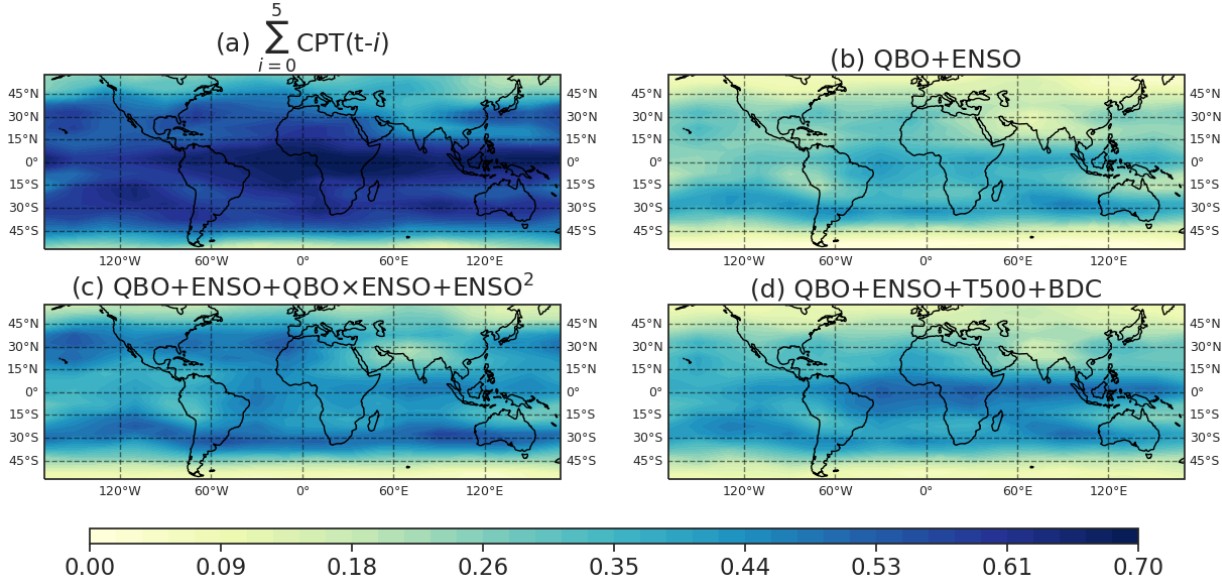

**Figure 3.** The $R^2$ of the MLR between water vapor anomalies at the 82.54 hPa level with the four groups of predictors: (a) cold point temperatures; (b) QBO and ENSO; (c) as in (b) but adding in $ENSO^2$ and QBO×ENSO; (d) as in (b) but adding in T500hPa and the BDC. This MLR spans from 2005 to 2019 and uses the 3D SWOOSH product. The regression is reconstructed directly from all predictors, i.e., in-sample.

(SVM), Random Forest (RF) and Multi Layered Perceptron (MLP). These ML models are applied in Section 3 only. All the models here are implemented through the Scikit-Learn Python package (Pedregosa et al., 2011) and use an optimization scheme in order to reduce the error between the predicted and the observed target variable. However, each of the models' approach to the regression task is different.

The SVM model, in a classification task, uses a linear hyperplane in order to separate each sample class (Boser et al., 1992). By applying the kernel trick, the input variables are non-linearly transformed into a high dimensional space where the type of the kernel, e.g., radial basis function, can be determined by hyper parameter tuning (Vapnik et al., 1995). In regression tasks, an error parameter is added ($\epsilon$) which measures the constraint on the residuals. The RF model uses many independent decision trees on randomized selections of the trained data subsets. The final output is produced by averaging all of the individual

decision tree outputs (Breiman, 2001). The MLP is a neural network algorithm that includes multi layered nodes with weights (Hinton, 1990). Typically, the network architecture includes an input layer, any number of hidden layer and an output layer where each layer's nodes are connected via activation functions (a so-called feed–forward propagation). During the learning process, the weights are re-evaluated using the back-propagation iterative algorithm (Orr and Müller, 2003) in order to decrease the cost function. Typically, the number of hidden layers in the MLP architecture is determined in the hyper-parameters tuning

step, and in our case was 1 hidden layer with 10 hidden units.



Finally, we use Multiple Linear Regression (MLR), a well-known and often-used technique in the field (e.g. Dessler et al., 2013; Diallo et al., 2018). When applied to lat-lon entry water vapor data, the model yields:

$$\chi_{\text{H2O}}(t, \phi, \lambda) = \alpha(\phi, \lambda) + \beta_i(\phi, \lambda) \cdot \eta_i(t) + \epsilon(t, \phi, \lambda) \tag{1}$$

where $\chi_{H2O}$ is the reconstructed water vapor anomalies field and $t, \phi, \lambda$ are the time, latitude, and longitude respectively. $\alpha$

and $\beta_i$ are the intercept and the beta coefficients of the MLR solution and $\epsilon$ is the residual field. $\eta$'s are the predictors used in the analysis. Note that this MLR is computed separately for each grid cell using the 3D SWOOSH data since 2005. We also perform an MLR using the tropical mean entry water since 1994, where we average the latitude range between 15°S and 15°N, and the predictors are QBO and ENSO. Thus a much simpler linear model is formulated as follows:

$$\chi_{\text{H2O}}(t) = \alpha + \beta_i \cdot \eta_i(t) + \epsilon(t) \tag{2}$$

The validation and testing procedures of the ML models is done in two stages using a 5 fold Cross Validation (CV) technique for each model separately. First, for the validation stage, we randomly select 80% of the samples and split them into 5 random groups called folds. Second, we train each model on 4 folds and test its performance ($R^2$) on the remaining fold. Third, we repeat this process 5 times (hence 5 fold CV) while iterating over all the folds. These three steps are repeated for all possible combinations of the hyper-parameters, and we then choose best hyper-parameters which maximize the out-of-sample $R^2$.

(This step is skipped for MLR since it does not have hyper parameters.) Then, for evaluating the models' performance, we traditionally would test the models once on the remaining data (i.e. the test set), however, since our dataset is quite short (312 samples at most), we would like to gain understanding of the models' performance distributions. Thus, we use a similar 5-fold CV on all the samples: we randomly divide the data in five, train each model on 4 folds, and test its performance ($R^2$) on the remaining fold; these steps are cycled through all five folds. This random division of the data into five folds and subsequent

cycling is performed 20 times, and so we end up with 100 $R^2$ scores per each model.

In the spirit of reproducible science, we encourage the interested reader to explore the Python repository hosted on GitHub.com (https://github.com/ZiskinZiv/Stratospheric_water_vapor_ML) that includes the processed data (except SWOOSH data sets) and procedures of this paper's analysis.

## 3 Re-evaluation of the importance of ENSO and the QBO in the observational record

We begin with the reconstructed entry water vapor timeseries in Figure 4 as computed by four different techniques with the QBO and ENSO used as predictors. As discussed in section 2.3, we use out-of-sample testing to reduce as much as possible overfitting. Specifically, Figure 4 shows the mean of the predicted out-of-sample water vapor from the 5-fold cross validation scheme (see section 2).





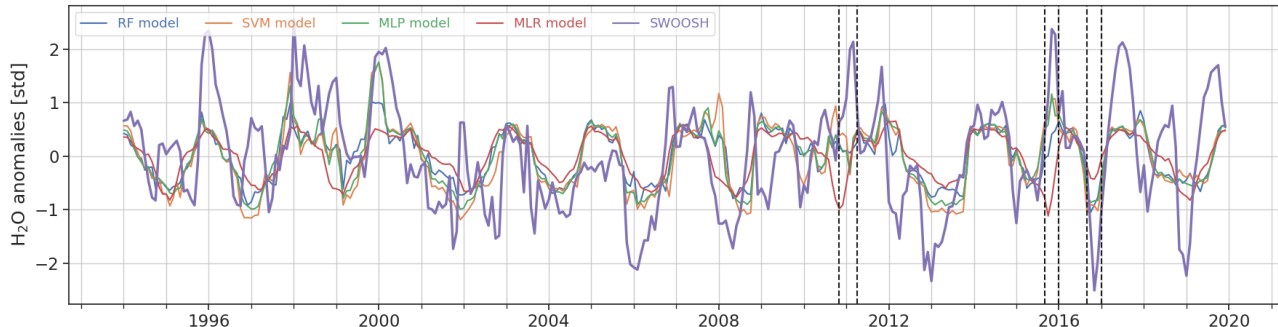

**Figure 4.** Out of sample model predictions of deseasonalized and standardized water vapor at 82.54 hPa averaged between 15°S and 15°N. The various models are RF (blue), SVM (orange), MLP (green) and MLR (red). The observations are from SWOOSH (purple). Note the three forecast "busts":2010-D-2011-JFM, 2015-OND and 2016-OND.

All four methods capture much of the variability of entry water present, but there are noticeably more forecasts busts than if cold point temperatures are used as in Randel and Park (2019). Three examples of forecasts busts are evident in late 2010, late 2015, and late 2016 (indicated by vertical lines), when all four techniques struggle to account for the observed change [1]

The ability of each of the four techniques is quantified in Figure 5a-d, which shows a histogram of the $R^2$ between the
predicted and actual entry water vapor for each of the individual out-of-sample tests performed. Figure 5 also indicates the mean, median, and standard deviation of the histogram of out-of-sample tests, and also the $R^2$ if we compute the fit using all data instead of applying an out-of-sample test. For all four techniques, there is a wide range of $R^2$ values among the 100 different out-of-sample tests, and the in-sample $R^2$ always exceeds the median of the 100 out-of-sample tests. This highlights the need to perform an out-of-sample test to minimize overfitting. If the 3 ML techniques are compared to MLR, the MLR is
the least successful both when applied in-sample and out-of-sample, and the three advanced ML techniques all are similarly skillful (with SVM slightly worse than MLP or RF).

This comparison of MLR in Figure 5d to the ML techniques in Figure 5a-c may lead to an underestimate of the abilities of MLR to account for entry water vapor, as the ML techniques allow for nonlinearity but MLR does not. As discussed in the introduction, there are at least two nonlinear processes that have been argued to exist when accounting for entry water vapor
variability due to ENSO and the QBO: $ENSO^2$ and a ENSO × QBO predictor. We therefore add these two predictors to the MLR, and repeat the calculation in Figure 5e. While the in-sample result is still lower than that of the ML techniques (likely because of additional nonlinear effects that are not included in the MLR), the out-of-sample results are now similar to those of the ML techniques. In other words, adding these two nonlinear processes can explain most of the additional advantage of the ML techniques when the data is tested out-of-sample to mitigate over-fitting.

---

[1]Note that the bust in late 2010 may be improved if the extension of the QBO to the lowermost stratosphere was taken into consideration (Davis et al., 2013), however the QBO in many CMIP models cannot be defined any lower than 50hPa.



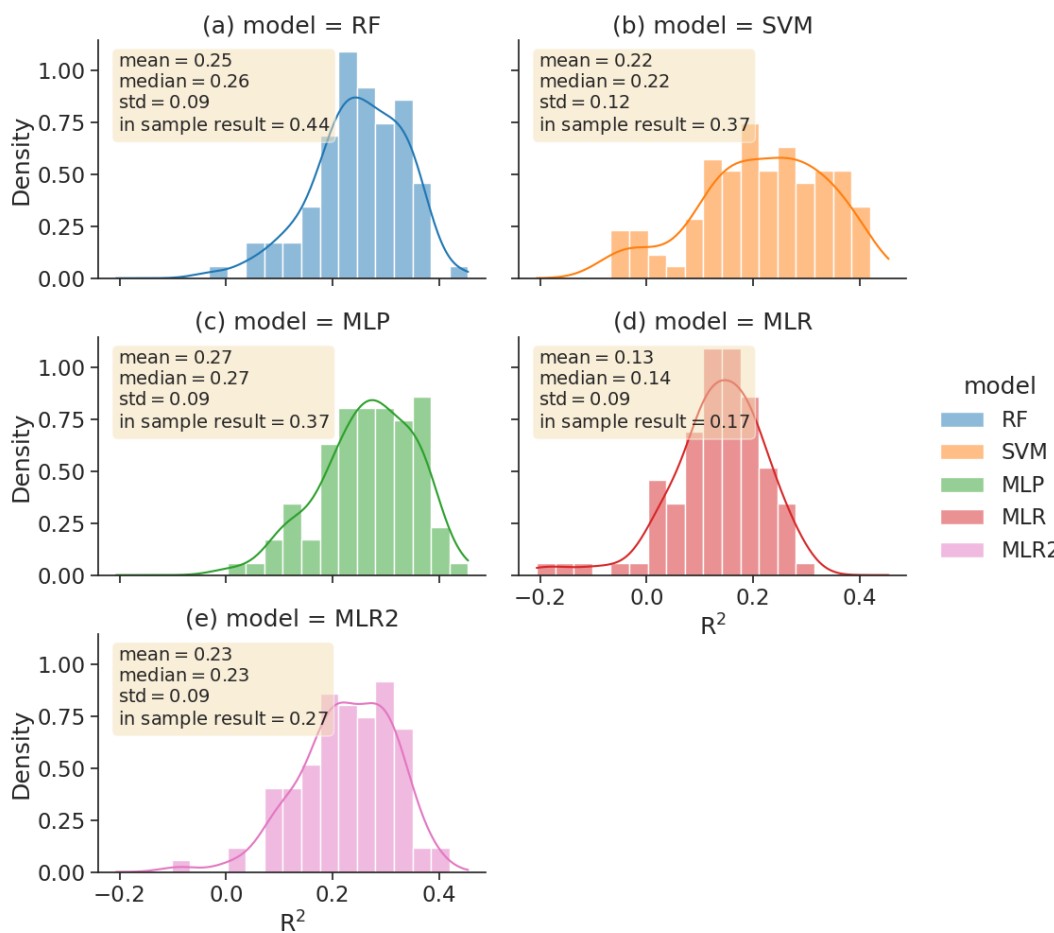

**Figure 5.** Out of sample model performance and distribution of $R^2$ scores of deseasonalized and standardized water vapor at 82 hPa averaged between 15°S and 15°N. The various models are RF (blue), SVM (orange), MLP (green) and MLR (red) and MLR2 (pink). The MLR2 model is the same as the MLR model but with $ENSO^2$ and $ENSO \times QBO$ predictors. The mean, median and std are noted for each distribution in a yellow text box, along with the in sample $R^2$ score.



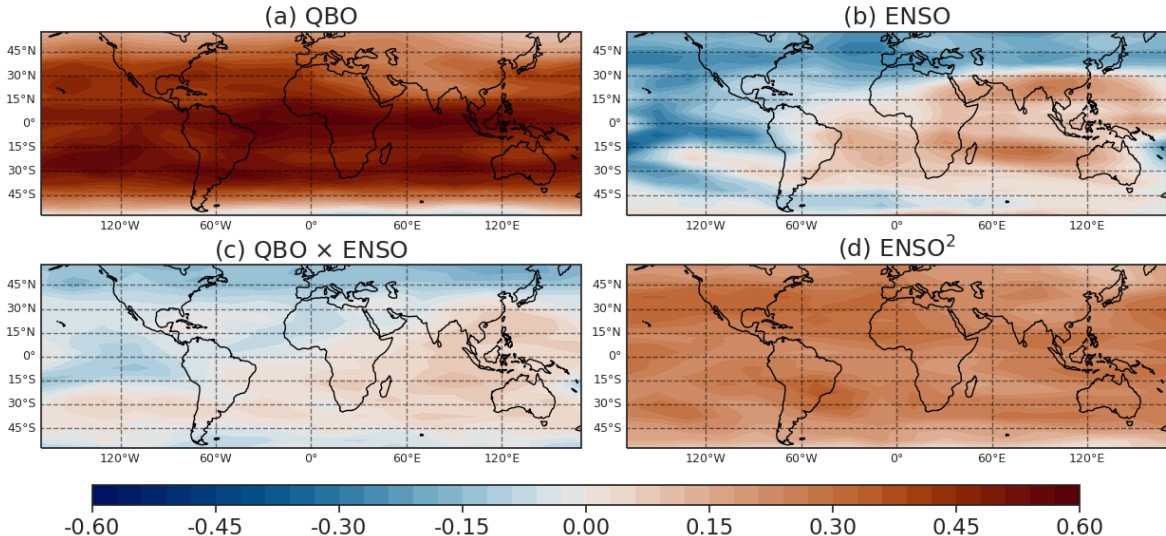

**Figure 6.** The in-sample $\beta$ coefficients for the water vapor anomalies MLR analysis in the 82.54 hPa level at 2005 to 2019, performed using the 3D SWOOSH data.

Even though these nonlinear processes help, the resulting $R^2$ is still much less than that explained by CPT (Figure 3a). Specifically, Figure 3b shows that a MLR with just QBO and ENSO can lead to an $R^2$ ranging around 0.3, however this is only half of the $R^2$ when the actual cold point temperatures are included (Figure 3a). Adding the two nonlinear predictors (Figure 3c) leads to an increase of $R^2$ by around 0.1 as compared to Figure 3b, but this is still much less than the $R^2$ in Figure 3a.

At least two of the techniques considered allow for a clear diagnosis of the relative importance of ENSO vs the QBO: MLR and SHapley Additive exPlanations (SHAP) as employed on the RF model. The relative importance of each of the predictors in the MLR of Figure 5e is shown in Figure 6, which shows a latitude vs longitude map of the regression coefficients when the regression is performed for water vapor at each gridpoint separately (the MLR of Figure 5e is performed on the tropical mean water vapor.) The QBO is clearly more important than any of the other processes for accounting for entry water, and thus

accounts for the biennial nature of the fit in Figure 4. Interestingly, the map for ENSO indicates a zonally asymmetric structure (Figure 6b), and as discussed in Yulaeva and Wallace (1994) and Garfinkel et al. (2013) the temperature structure of ENSO is characterized by zonal structure even in the lower stratosphere, with relatively warm temperatures in the Indian Ocean sector and colder temperatures in the Pacific sector. This zonal temperature dipole thus is apparently leading to a similar dipole in entry water, with moistening occurring in warm regions and drying in cold regions. The $ENSO^2$ predictor is more important

than ENSO for zonal mean entry water vapor (Figure 6bd). The ENSO times QBO predictor is comparatively unimportant (Figure 6c).

The SHAP technique also allows for quantification of the relative impact of ENSO versus QBO. SHAP (Lundberg et al., 2020) implements a concept borrowed from game theory, where a prediction can be explained by assuming that each predictor's





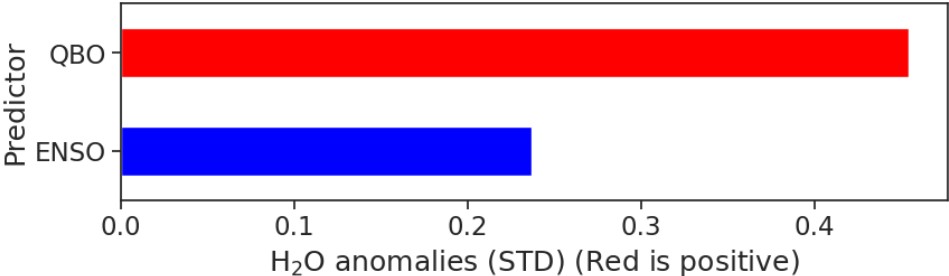

**Figure 7.** Mean SHAP values for the RF model as reconstructed in Fig. 4. Red (Blue) color indicates that the predictor is positively (negatively) correlated with the target variable, i.e., water vapor anomalies.

value of the instance is a "player" in a game where the prediction is the payout. The Shapley values (as computed by e.g., SHAP) indicate how to fairly distribute the "payout" among the predictors (Lundberg and Lee, 2017). The "payout" in our problem is the standardized entry water, thus the computed Shapley values are measuring the mean effect ENSO or QBO have on standardized $H_2O$ anomalies. For an in-depth explanation of the SHAP technique, we encourage the interested reader to

explore the SHAP chapter of the online book on Explainable AI (Molnar, 2019) methods.

Fig. 7 shows the mean SHAP values for the predictors as trained by the RF model. QBO is twice as important as ENSO where QBO is positively correlated with $H_2O$ anomalies while ENSO is negatively correlated. Only when considering spring entry water is ENSO positively correlated, (Garfinkel et al., 2018) though even in spring the QBO dominates.

Additional evidence as to the importance of the $ENSO^2$ predictor is provided in Figure 8, where we form an MLR using

QBO and ENSO but compute the ENSO regression coefficient separately for each ENSO phase. The important point is that the regression coefficient changes sign between EN and LN (Figure 8b vs. Figure 8c); in other words, a more positive ENSO state during EN leads to more water vapor, but so does a more negative ENSO state during LN. A naive MLR misses this effect, and would imply a limited impact of ENSO on entry water vapor. Only upon considering nonlinear effects is the full impact of ENSO revealed.

Finally, some previous work has focused on using the BDC or mid-tropospheric temperatures as predictors in MLR models that attempt to explain entry water (e.g., Dessler et al., 2014). We show the $R^2$ of an MLR with these predictors in Figure 3d. Adding T500 and the BDC clearly leads to an improved fit as compared to an MLR with only QBO and ENSO ( Figure 3b vs. Figure 3d), however the improvement is similar to the effect of the nonlinear regressors in Figure 3c. As discussed in section 2, there is a significant correlation between the BDC at 70hPa and the QBO at 50hPa, and hence including both in a ML

model does not lead to significant improvement. Including the BDC at 82hPa instead leads to a larger improvement, however the BDC at 82hPa is significantly correlated with the cold point temperatures, and hence there is ambiguity if the BDC is defined at 82hPa instead. There is some added value to using T500 as compared to ENSO, though as shown in Figure 6, an



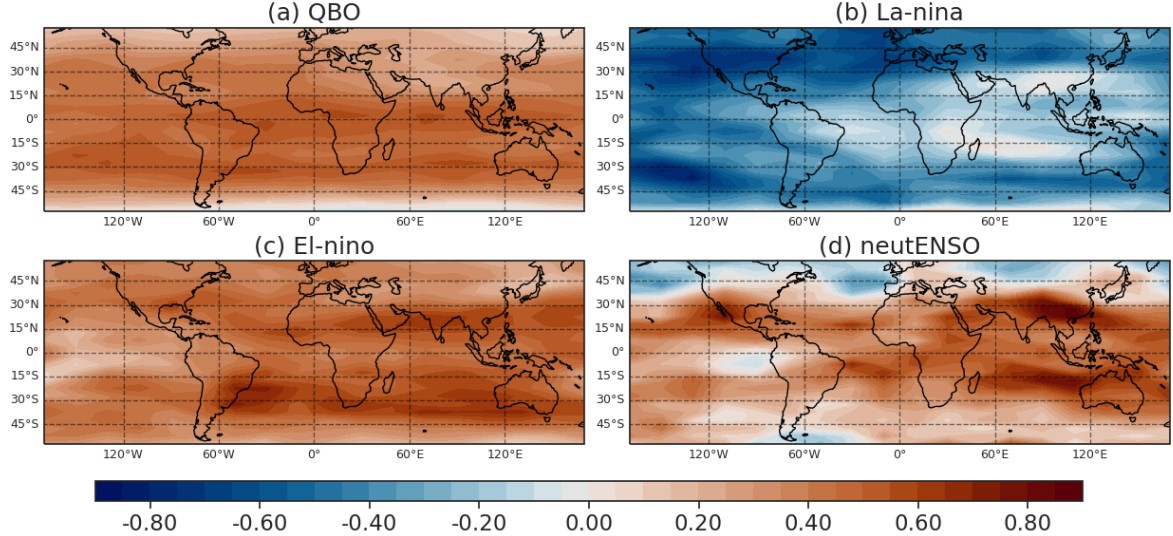

**Figure 8.** The in-sample $\beta$ coefficients for the water vapor anomalies MLR analysis in the 82.54 hPa level at 2005 to 2019. The ENSO predictor was separated into 3 parts where EN represents the El-nino events (ENSO>=0.5), LN represents the La-nina events (ENSO<=-0.5) and neutENSO the rest of the ENSO regressor.

ENSO predictor is much less useful than an $ENSO^2$ predictor in any event. That is, most of the improvement upon adding the nonlinear predictors comes about via the $ENSO^2$ predictor (Garfinkel et al., 2018).

## 4  Ability of CMIP6 and CCMI models to represent the QBO modulation

Section 3, and specifically Figure 6, indicated that the QBO is the most important single predictor of any considered in this
paper barring the cold point temperatures themselves. We now consider the ability of CMIP6 and CCMI models to represent this connection, and for simplicity we focus on a simple regression of the QBO with entry water. (The ability of these models to represent the connection between ENSO and entry water vapor was considered in Garfinkel et al. (2021) in detail.)

The lagged correlation of the QBO with entry water is shown in Figure 1a for the CCMI models, and in Figure 1b for the CMIP6 models. While all models capture the sign of the dependence of entry water on the QBO (an apparent improvement
from Smalley et al., 2017), there is a wide range in the amplitude of the correlation. The two NCAR models in CCMI simulate the strongest relationship, but these models nudge their QBO, and the corresponding CMIP6 run with a spontaneous QBO simulates a weaker connection. Other models simulate a connection similar to (HadGEM3, EMAC-L47MA), or weaker than (NIWA-UKCA, MRI-ESM), that observed. Note that Smalley et al. (2017) considered these latter two CCMI models and also found a nearly non-existent connection between entry water and the QBO.
An alternate perspective on the ability of models to capture the relationship between the QBO and entry water is the regression coefficient from a MLR. Figure 9a shows these regression coefficients. The two NCAR models in CCMI are the only





models with a regression coefficient approaching that observed (the horizontal black line). The other models uniformly underestimate the regression coefficients, and hence the relatively more realistic correlation coefficients from Figure 1 (which are repeated in Figure 9c) are due to biases either in the standard deviation of entry water vapor or in the standard deviation of the QBO itself. Garfinkel et al. (2021) already demonstrated that ten of these models (with NIWA the lone exception) underesti-

mate entry water variability. These models also mostly underestimate variability of the QBO, as shown in Figure 9b. While e.g., the UK Met Office model does a good job at capturing the QBO (and recall the NCAR CCMI models have a nudged QBO), most other models struggle. A notable improvement is evident from the MRI contribution to CCMI to the MRI contribution to CMIP6. The net effect of too weak internal variability of the QBO or of entry water is that the regression coefficient of a model will be lower than that in observations, even if the correlation is generally realistic.

Do models with a better QBO perform better at capturing the relationship between entry water and the QBO? Figure 9d compares for each model the standard deviation of the QBO (x-axis) with the correlation between entry water and the QBO (y-axis), and it is evident that the two are linked. The correlation coefficient across all models is statistically significant at the 95% level. Hence, an improved QBO leads to an improved representation of interannual variability of entry water.

## 5   Discussion

Stratospheric water vapor plays a crucial role both as a greenhouse gas that modulates the Earth's radiative budget, and also as a trace gas that regulates the severity of ozone depletion (Solomon et al., 1986; Forster and Shine, 1999; Solomon et al., 2010; Dessler et al., 2013; Wang et al., 2017; Banerjee et al., 2019). This study aims to understand the importance of nonlinearity for two processes - ENSO and the QBO - that have been shown to regulate water vapor concentrations on interannual timescales, and to consider whether comprehensive models used for climate change assessments represent these factors correctly.

Both the QBO and ENSO are important for entry water vapor, however a simple linear perspective would lead to the mistaken conclusion that the effect of ENSO on zonal mean entry water vapor is minimal (Figure 6b). Rather, $ENSO^2$ is the more important contributor (Figure 6d), though even $ENSO^2$ is less important than the QBO (Figure 6a). A multiple linear regression model that includes ENSO and the QBO performs notably worse than machine learning techniques that do not assume linearity (Figure 5a-d), however adding an $ENSO^2$ predictor to a multiple linear regression model fills the gap in

performance (Figure 5e), and the added value from the more complicated machine learning techniques is small.

    Most of the comprehensive models considered here underestimate the strength of the connection between the QBO and entry water vapor (Figure 1 and 9), with the only exception models which nudge the QBO rather than spontaneously generate it. While this result is disappointing, a notable improvement is evident from the CCMVal2 and the early CCMI data analyzed by Smalley et al. (2017). We find that models in which the QBO reaches the lower stratosphere tend to perform better at

capturing the relationship between entry water and the QBO (consistent with Geller et al., 2016), and QBO propagation into the lowermost stratosphere is crucial also for QBO teleconnections to the subtropical jet, to the Arctic stratosphere, and to tropical convection (Garfinkel and Hartmann, 2011; Garfinkel et al., 2012; Martin et al., 2021).





**Figure 9.** Relationship between the QBO and entry water vapor in CCMI and CMIP6 models. (a) regression coefficient; (b) standard deviation of the QBO at 50hPa; (c) correlation coefficient; (d) relationsip between the correlation coefficient (panel c) and standard deviation (panel b). For (d), diamonds are CCMI models and stars are CMIP6 models.





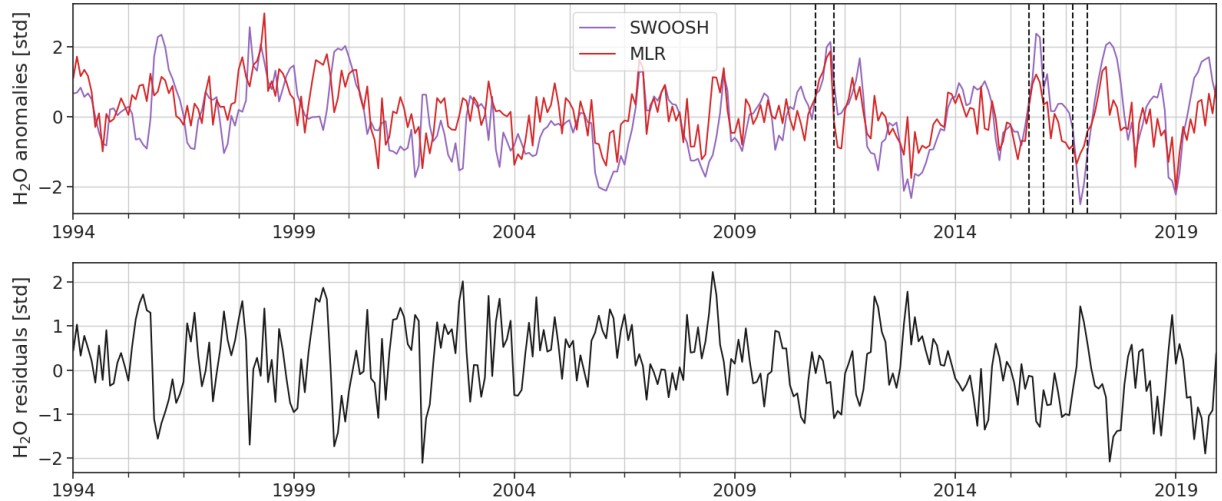

**Figure 10.** Deseasonalized and standardized water vapor at 82.54 hPa averaged between $15°$S and $15°$N (purple) and their MLR reconstruction (red) and residuals, spanning from 1994 to 2019. This MLR analysis was carried out with the Randel and Park (2019) CPT as the only predictor but after detrending the data. The MLR model was trained on the MLS portion of SWOOSH (2005 to 2019, correlation=0.68) and was reconstructed on the full time span (1994-2019, correlation=0.59).

When considering the total variance of entry water vapor in Figure 5, the out-of-sample $R^2$ was always less than the in-sample $R^2$. The importance of out-of-sample testing is further illustrated in Figure 10. Figure 10a shows the timeseries of zonal mean water vapor from SWOOSH and the MLR reconstruction if the detrended Randel and Park (2019) CPT is used as the sole predictor for detrended entry water and the model is trained over the period 2005 to 2019 only. While the MLR model

does a reasonable job of explaining the observed variability over the period used for training the MLR model, the MLR fails when applied out-of-sample to the pre-MLS period (Figure 10b), as reflected by the generally larger values of the residuals. In other words, the model is overfit to the training data, and is not generalizing well to out-of-sample data. This kind of overfitting can be minimized by appropriately tuning the hyperparameters for the ML techniques, though for MLR the only remedy is to perform out-of-sample testing. Hence we strongly recommend that future studies using MLR or similar techniques use some

variant of out-of-sample testing to minimize over-fitting.

While the ENSO predictor is only weakly related to zonal mean entry water vapor, ENSO is associated with zonal structure in water vapor in the lower stratosphere. Figure 6b shows that water vapor is enhanced over the Indian Ocean sector, and reduced over the East Pacific sector. This zonal dipole resembles the zonal dipole of temperature in the TTL (e.g. Garfinkel et al., 2013, 2018), with locally warm TTL conditions associated with moistening and locally cold TTL conditions associated

with drying. Note that higher in the stratosphere, this zonal dipole goes away. However this results suggests that up to 82hPa, horizontal motions are still not fast enough to fully homogenize tropical water vapor, as might be expected if the tape recorder mechanism was the only relevant mechanism (Mote et al., 1996b). Future work should consider whether other factors (e.g.,

SST patterns not related to ENSO) may also lead to zonal structure of water vapor in the lower stratosphere. Future work should also consider additional novel means of interpreting the improvements of the ML fits as compared to MLR, in order to bridge the gap between an improved fit and an understanding of how and why the improvement came about.

Finally, cold point temperatures (CPT) control around 75% of the variance of entry water vapor over the historical record.
None of the large scale predictors, neither individually nor in combination, come even close to explaining such a large fraction of the variance (Figure 3). This gap in explainable variance highlights the need to better understand CPT variability on interannual timescales, and perhaps even to build predictive models for the CPT itself.

*Code and data availability.* The CCMI model output was retrieved from the Centre for Environmental Data Analysis (CEDA), the Natural Environment Research Council's Data Repository for Atmospheric Science and Earth Observation (https://data.ceda.ac.uk/badc/wcrp-
10 ccmi/data/CCMI-1/output) and NCAR's Climate Data Gateway (https://www.earthsystemgrid.org/project/CCMI1.html). All the non-linear ML models and the MLRs are implemented through the Scikit-Learn Python package (Pedregosa et al., 2011)

*Author contributions.* SZZ conceived of and performed the machine learning analysis, and led the analysis. CIG performed the CMIP6/CCMI analysis. AB provided feedback on the ML methods. SZZ and CIG wrote the paper jointly, and feedback was provided by SD and AB. SD also contributed the observational water vapor data.

*Competing interests.* The authors declare that they have no conflict of interest.

*Acknowledgements.* We thank the international modeling groups for making their simulations available for this analysis, the joint WCRP SPARC/IGAC CCMI for organizing and coordinating the model data analysis activity, and the British Atmospheric Data Centre (BADC) for collecting and archiving the CCMI model output. Correspondence should be addressed to Chaim I. Garfinkel (email: chaim.garfinkel@mail.huji.ac.il).



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
