# Peer review of "The roles of the Quasi-Biennial Oscillation and El Nino for entry stratospheric water vapour in observations and coupled chemistry-ocean CCMI and CMIP6 models"

_Atmospheric Chemistry and Physics, 2021_

## Author Comment (AC1)

**Answer to Reviewer #1**

We would like to start by thanking you for all the time and effort which you spent reviewing our paper. All your comments, suggestions, and questions were taken into account and all the necessary corrections were made in the revised manuscript. Furthermore, we address all your comments and suggestions below, point by point.

**General comments**:
The paper focuses on the factors affecting the interannual variability of stratospheric water vapor entry in the tropics in observations, CCMI and CMIP6 models. The authors contrast the use of a variety of techniques: multiple linear regression and 3 machine learning methods. Cold point temperatures are the main factor explaining the water vapor variability. They discuss the merits of the different techniques and the relative importance of the QBO and ENSO. They also find non-linear interactions to be important. The comprehensive models, whilst will suffering from a QBO that is not deep enough, have nonetheless improved. The paper is well written and provides an good description of machine learning techniques applied to a geophysical problem. The figures are also mostly clear.

**Specific comments**:

(1) Make it clear earlier during the introduction that you are looking at interannual variability and not the seasonal cycle.

We clarified this in the introduction.

(2) Some of the CCMI model have multiple ensembles. Do you average over all of them? If so, does this result in less variability and thus make it harder to compare to those runs with only 1 ensemble?

We include each ensemble member separately. We don't average the ensembles together before computing correlations, rather compute for each ensemble member separately. Now clarified.

(3) In the figures, would it be possible to have the models with a nudged QBO labeled in bold text? It would make identifying them easier.

We tried adding this information to the figure legend of figure 1 and 9, but the figures then looked strange. We added it to the caption instead.

(4) On line 4, page 6, you mean ERA5/ERA5.1 I think?

Yes, we corrected this.

(5) On page 6, line 11, "Note that the correlation of the BDC with the QBO is -0.66 (Figure 2), and hence including both in a single regression or ML model can lead to overfitting. " I disagree with this statement. Multicollinearity in your predictors causes a variety of problems but does not specifically cause overfitting. See page 283, Applied linear statistical models 5th edition by Neter et al. (2004). Your validation stage should show if overfitting is an issue.

Indeed, we rectified the sentence to say that multicollinearity can lead to erroneous model interpretation.

(6) Page 10, line 15, the non-linear predictors are interesting but I struggle to relate them to physical processes. Could you give the reader a sense of what $ENSO^2$ might be?

Garfinkel et al 2018 goes into great detail as to why physically La Nina can also lead to a moistening. The short answer is that the region of the cold point moves zonally within the tropics, and even though the lower stratosphere cools, the cold point actually warms.

This has been added to the introduction section where it seems more appropriate than at this point in the text:
"Both  La Nina and El Nino can lead to a moistening if the  cold point moves zonally within the tropics (to the Central Pacific for El Nino, and to the far West Pacific for La Nina), and even though the lower stratospheric response is opposite for El Nino and La Nina, the cold point warms for both (Garfinkel et al 2018)" We also added a similar sentence to the discussion.

(7) The values in Figure 6 are somewhat hard to read. Could you add a few labeled contour lines please?

Contour lines with labels were added to Fig. 6, Fig. 3 and Fig. 8.

(8) Figure 7 feels unnecessary since the same information can be conveyed with the text.

We removed Fig. 7 from the paper and updated the text to include the SHAP values for each predictor.

(9) In figure 9 (a to c), the text suggests that the solid black lines are observations (and they are not described in the caption) but where are there two parts and at different values? Label the models in 9(a).

We now note the solid black horizontal line is observations, and that entry water is defined separately for CCMI and CMIP (80hPa and 70hPa respectively).

Adding labeling to panel 9a made the figure more visually distracting without any added content, hence we left 9a as is.

**Minor comments**

Page 1, line 164, Emissions

corrected

Page2, line 5, through the its

corrected

Figure 1. Labels are a bit small and hard to read.

now larger

Figure 4. Are the units of the H20 anomalies correct?

Yes, we clarified it in the caption.

Figure 5 and Figure 9. You use "std" and "std dev". Choose one to be consistent and also explain the abbreviation in the caption.

We have adopted the std.dev abbreviation and updated it in the caption.

Figure 5(a) I am confused about the histogram. Is it normalized? If so, why are the values >1?

The histogram is normalized in a way that the total area of the histogram equals 1. This means that some bars can indeed exceed 1. However, it may be confusing, thus, we replaced the figure with the "probability" normalization where the sum of all the bars equals 1. This changes only the y-axis values and not the shape of the histogram which is more important in the context of this paper.

---

## Author Comment (AC2)

**Answer to Reviewer #2**

We would like to start by thanking you for all the time and effort which you spent reviewing our paper. All your comments, suggestions, and questions were taken into account and all the necessary corrections were made in the revised manuscript. Furthermore, we address all your comments and suggestions below, point by point.

**General comments**:
This paper discusses the importance of the nonlinear interaction between ESNO, QBO, and stratospheric water vapor, based on MLR and advanced machine learning techniques, and analyzes both observational data and chemistry-climate models. The authors conclude that QBO is more important than ENSO^2 than ENSO in predicting entry water vapor. The novel techniques and rigorous analysis of this paper will inspire the whole community, and I recommend this paper be accepted after a few revisions.

1. As the authors mentioned in line 5 and line 13 page 2, ENSO and QBO influences the stratospheric water vapor by influencing the tropical tropopause temperature. Later in Fig. 3, the authors compare the prediction of water vapor from merely tropical tropopause temperature, and from linear/nonlinear combination of ENSO and QBO. Since the ENSO and QBO directly influence tropical tropopause temperature and indirectly influences water vapor, before showing the relationship between 'ENSO, QBO-stratospheric water vapor', additional analysis of how well can linear/nonlinear combination of ENSO and QBO represents the tropical tropopause temperature will make the logic tighter.

Garfinkel et al 2018 and 2021 considered the influence of ENSO on tropical tropopause temperatures in great detail, and we have nothing to add here. We have added more discussion of these papers in the introduction and discussion sections.

The role of the QBO for tropopause temperatures has also been considered extensively in previous work of others, including the papers we cite (Reid and Gage, 1985; Zhou et al., 2001, 2004; Fujiwara et al., 2010; Liang et al., 2011; Kawatani et al., 2014). We don't have much to add here either. The connection is known theoretically (as given by thermal wind balance on an equatorial beta-plane) to be linear.

2. It is undoubted that considering the nonlinear process from ENSO and QBO can substantially increase the prediction of stratospheric water vapor, from the statistical analysis of this paper. However, more scientific arguments are needed when showing this result. For example, ENSO^2. The difference between ENSO and ENSO^2 are (1) ENSO^2 always amplifies extreme positive and negative ENSO states; (2) ENSO index has positive and negative values, but ENSO^2 only have magnitude, so extreme EN and LN will have similar ENSO^2 values. The

authors explain (2) in section 3, but lack the necessary analysis of how (1) influences the predictions. Can you add another experiment of, say, absolute(ENSO)? It is possible that the behavior of abs(ENSO) is not as good as ENSO^2, since moderate events are not very important and ENSO^2 emphasizes the importance of extreme events so not necessary to add this experiment into the paper. Then I suggest that can add some more comments on page 13, lines 9-14 on how the two differences between ENSO and ENSO^2 improve the prediction. I also suggest including citations of why choosing ENSO^2 and ENSO*QBO not only in the introductions but also in result sections when discussing the improvement.

Garfinkel et al 2018 goes into great detail as to why physically La Nina can also lead to a moistening. The short answer is that the region of the cold point moves zonally within the tropics, and even though the lower stratosphere cools, the cold point actually warms. This has been added to the introduction section and discussion section.

Regarding your comment concerning ENSO^2 vs. abs(ENSO), below we copy in relevant figures from Garfinkel et al 2018 showing a scatter plot of the entry water vapor values for different values of the Nino3.4 index.

for the GEOSCCM model:

[Figure]

for the observations:

[Figure]

While one could attempt to discriminate whether abs(ENSO) is "better" than ENSO^2, there simply aren't enough points to make a convincing statistical case either way.

**Specific comments**:

1. In figures showing the horizontal distributions, i.e., Fig.3, Fig.6, and Fig. 8, since ENSO is one of the most important topics of this paper, I suggest the base map should center at 180° instead of 0°, so the readers can compare the Western and Eastern Pacific more clearly.

We changed the center of these Figs to 180°.

2. 10, please add panel numbers and titles.

We added the panel designations and titles.

3. Page 1, line 15: please include more citations for 'The amount of water vapor that enters the stratosphere is also important for stratospheric chemistry and specifically the severity of ozone depletion, for example, the citations on page 15, line 17.

We added three more.

4. Page 4, line 21: 'In total, more than 2500 year of model output are available' I see no reason to calculate the total years because you are not putting all the model outputs together.

This sentence has been removed.

5. Page 6, line 8: please introduce more about the radiosonde data, for example, is it monthly mean? Is the seasonal cycle included?

The radiosonde data was resampled to monthly means and its seasonal cycle was removed. We clarified this in the text.

6. Page 9, line 22: thanks for sharing, this is helpful to the community!

You're welcome :-)

7. Page 10, line 15: is the 'busts' problem in figure 4 still there in MLR2? 2010, 2015, and 2016 are all ENSO active years or right after so it is interesting to see whether adding ENSO^2 and QBO*ENSO can improve the performance or not.

These busts are present for MLR2 as well, though the error is not any larger than for the ML methods (we added MLR2 to Fig. 4).

8. Page 17, line 15: 'this results' should be 'this result'

corrected

---

## Referee Report (RR1)

**"The roles of the Quasi-biennial Oscillation and El Nino for entry stratospheric water vapour in observations and coupled chemistry-ocean CCMI and CMIP6 models" by Shlomi Ziskin Ziv et al. submitted to ACP**

**1. Summary**

Ziv et al investigate the roles of the QBO and ENSO for the interannual variability in entry stratospheric water vapour in observations and chemistry climate models outputs from CCMI-1 and CMIP6 using their Multiple linear regression and different supervise learning regressions (named here Machine learning). They compare the ability of their own MLR with the different supervise learning regressions to evaluate their MLR robustness compared to their different supervise learning regressions in capturing the interplay between the QBO and ENSO influences to water vapor entry.

The results idea is of great interest to potential readers and worth it for publication. However, the manuscript has 3 mains issues, which are a lack of honest motivation of the study, methodological failure and finally the presentation of the result issue that I have detailed in the major and specific comments. Major revisions are needed in order to make the paper suitable for publication. There are some additional points that need to be clarified. I apologize if I misunderstood something.

**2. Major comments**

i.  *Honest motivation of the study*: The concern is the main motivation of the paper. From Page 2, L13-35 and Page 3, L1-10, the discussion is not clearly and honestly reported. The authors are using the result from Garfinkel et al. 2018 based on CCMs (geosccm) to argue about questionable "nonlinear" ENSO response. The result of "nonlinear" ENSO impact on water vapor is still an open question as the observations does not show this nonlinearity. In addition, the models used in Garfinkel et al 2018, 2021 are based on spontaneous generated QBO, which does not reach the tropopause and does not have the same QBO phases and strength as the observations. This lead to wrong water vapor modulations by the interplay between the QBO and ENSO in the lower stratosphere, therefore, to questionable result of Garfinkel et al 2018, 2021. The discussion about about "El Nino and La Nina can lead to moistening" is still very questionable, knowing the inability of CCMs to reproduce a descent tape recorder (Keeble et al 2020). The look like circling discussion (e.g. dog biting its tail) about the ''nonlinearity'' of ENSO need to be discussed clearly as it still needs to be proven with the observations rather than it's already an established results. As we know CCMs have several issues when it comes to water vapor entry in the lower stratosphere. Finally, one important remark is the Authors are using their OWN multiple regression model, which is not the same as the Dessler et al. 2014, Diallo et al, 2018 and Tian et al. 2019. There are as many as different MLR in terms of predictors, including a dynamical or fixed lag and solver. Just note that the Diallo et al. 2018 method is not a simple MLR where you can predefine a fixed lag as your regression but a multivariate hybrid method. In order word you have used your OWN regression, therefore, you should be that general as even your regression has issues.

ii.  *Methodological failure:* The regression model used here is failing to reproduce the ENSO (El Nino and La Noina as well) induced impact on water vapor variability (Figure 6). The QBO coefficient also looks strange. The ENSO-square even looks like a second QBO coeficient. Actually, the ENSO impact on H2O structure is a horseshoe pattern as shown in Konopka et al 2016 and Avery et al. 2017. Apparently, the regression model used here only is not capturing ENSO look-like impact on H2O entry. When multiplying QBO impact by

ENSO impact (QBOxENSO), the result looks like an ENSO impact on H2O pattern. The Figure 7 and the large residual over the entire period (trained and tested) in Figure 9 both corroborate the failure of Ziv et al MLR model. The QBO signal from their MLR is also questioning. So, major analysis are need to investigate this failure before concluding. Possible diagnostics are: First, it would be great to see how well your MLR and ML are able to capture the altitude-time cross section of the tropical H2O variability induced by the QBO and ENSO (5S-5N mean of their effect). Second, estimate the R-square error of the residual. Third, verification of the used ENSO proxy if it is not too small and also especially in the manuscript (Page 7, L2), you stated using NINO3.4 from ERSSTv5 data with a 1981-2010, while the analysis period is 1994-2019. Regarding the ML, the different supervise learning method are barely described in the manuscript. The other main issue is the training and testing period of the Machine learning. The authors did not use an independent training date set for testing the performance of the machine leaning model. The approach of using the same data randomly sampled and divided into 5 fold won't help to assess the performance of the ML model. This is a serious issue. You should show at least show the ML performs in the unseen test sample to disclose over-fitting issues etc. The lag used in the manuscript is not clear if it is observed one or the one from the CCMs. Please clearly describe each method and explain what you have done. Finally, the cold point temperatures are very well negatively correlated with the H2O as the latter is determined by freeze-drying process (Fueglistaler & Haynes, 2005; Fueglistaler et al. 2013; Poshyvailo et al 2016; Grandville & Birner al 2016). The CPT as H2O then are both modulated by the climate modes of natural variability, including QBO and ENSO. So comparing CPT and the QBO and ENSO as predictors is not making sense at all. Since early findings, we know the strict relationship between H2O and CPT. One should use the CPT if one would like to predict H2O entry but when it comes to separating and understanding different contributions to H2O inter-annual variability, it does not make sense.

iii. *Presentation of the result issue*: The structure and presentation of the results have issues which need to be improved. The authors discussed about CCM2 (Page 4, L14-18), while they are not using it. I recommend to remove this part but clarify the CCM1 representation of QBO (nudged or spontaneous) and SSTs (modelled or observed), which missing here. For instance, EMAC has also the nudged QBO, which is not mentioned, but you emphasise the WACCM water vapor coefficient are due to the nudged. So it should be the same for EMAC bot no. In addition, the level of 82.54hPa used here is not a reference level, knowing that model like WACCM has a high tropopause (about 90 hPa). I would recommend to do these analysis of the manuscript at one fixed level 70 hpa for all data sets, which is actually the reference level where tropospheric influence is separated from the stratospheric ones. They could interpolate all the data at the 70hpa level.

**3. Specific comments**

a) Page 2, L9, Please add citations: Punge et al. 2009, Niwano et al. 2003 & Diallo et al. 2018.
b) Page 2, L13-20, please discuss the zonal mean struture of the ENSO induced impact on H2O based on the observation that has been found in previous litterature (Randel et al. 2009, Calvo et al 2010, konopka et al 2016). This is what so far the truth.
c) Page 2, L21-30 please rephrase the entire paragraph. The claimed ``nonlinear ENSO impact on H2O'' still need to be proved in the observations, therefore, it should not be presented as ground true the same models are pointed out having issues with the QBO, which stuck at 50hPa, not realistic QBO phases compare to observed one. Conclusions from these that struggle to reproduce the tape record should be take with caution, which is not the case here.
d) Page 2, L34-35 please remove the citations "Diallo et al. 218; and Tian et al. 2019" as they are not simpl MLR as you frame here.

e) Page 3, L3, this statement "First, Garfinkel et al 2018 found ...ENSO is nonlinear" needs to rephrase and made clear by precising it is model based and not consistence with the observations finding yet.

f) Page 3, L1-10, please discuss also these papers: Evans et al 2014; Brinkop et al 2016, Less et al 2012, Diallo et al. 2018 about the interplay between the ENSO and QBO impact on H2O entry.

g) Page 4, L1-10 please precise that you are using the CCMI phase 1 models. In addition, please explain the model issues about getting the QBO right in the CCMI-1 and CMIP6 models.

h) Page 4, L14-27 please remove the CCM-2 discussion. It is confusing the reader as any way you are focussing on CCMI-1. Please emphasize the models ability in reprodcuing ENSO and QBO impact on the tape recoder and the uncertainty that induces in the H2O entry.

i) Page 5, L1: Please do the analysis at 70hPha for all the plots.

j) Page 5, table 2, please the QBO and SST infor mation for each model in the table.

k) Page 6, the captions of Figure 1 are not very clear. Please clarify them.

l) Page 6, the Figure 1 should be done at 70hPa for all models and observation.

m) Page 7, L1-2, please clarify "...ERSSTv5 data with a 1981-2010 base period".

n) Page 8, L5-15, a clear description of the different supervised learning regression are need here to improve clarity of the method.

o) Page 9, L3-4, please remove the citations Dessler et al 2014 and Diallo et al. 2018 as you are not using their models or out put of their models for comparison. In addition, your regression model has issues in reproducing the ENSO and potentially QBO impact structure on H2O (Figure 6); tape recorder plot of QBO and ENSO induced impact on H2O and has large residual too.

p) Page 10, L1-8, the approach used here to test the performance of the model is an issue as it you're not test the ML on unseen data for test set. How the overfitting or under fitting issues are evaluated then? It would be great to add a figure in the main paper or supplement about the ML performance showing trained period and unseen predicted H2O period. Please clarify also the training period.

q) Page 11, L13-24, Here the authors should not generalise about the MLR and its results but precise it is THEIR MLR with its limitations. The whole paragraph nee to be revise after evaluating the ability of their MLR to capture QBO and ENSO induced impact on H2O as altitude-time tropical cross-section.

r) Page 11, L25-34, the SHAP method comes out off blue. Please clarify and rephrase the paragraph

s) Page 13, the coefficient of their MLR in Figure 6a & b are wrong as well as the Figure 7. ENSO impact on H2O is not similar to classical method results. Please evaluate clearly, why? In addition, the Figure 6 d e.g. ENSO squarred is very likely a QBO signal as you are not using two QBO index with a chosen lag for all latitude bin this may impact you MLR results. The MLR needs to be evaluated before drawing any useful conclusion here.

t) Page 13, L1-8, QBO being predominate in modulating H2O entry have been already found by Diallo et al. 2018 and confirmed by Tian et al 2019. Please discuss them.

u) Page 14, L2-5, knowing the model inability of reproducing the QBO down to the low stratosphere, it is a bit strange that the author aims at evaluating the model ability to capture the interplay between QBO and ENSO impact on H2O entry. Please rephrase the entences.

v) Page 17, L16, the zonal structure temperature and H2O anomalies find in previous studies (Randel et al 2009, konopka et al 2016) is a result of the averaged between a region of updraft (cold) and subsidence (warm).

---

## Author Response (AR2)

We would like to start by thanking you for all the time and effort which you spent reviewing our paper. All your comments, suggestions, and questions were taken into account and all the necessary corrections were made in the revised manuscript.
Furthermore, we address all your comments and suggestions below, point by point.

**1. Summary**
Ziv et al investigate the roles of the QBO and ENSO for the interannual variability in entry stratospheric water vapour in observations and chemistry climate models outputs from CCMI-1 and CMIP6 using their Multiple linear regression and different supervise learning regressions (named here Machine learning). They compare the ability of their own MLR with the different supervise learning regressions to evaluate their MLR robustness compared to their different supervise learning regressions in capturing the interplay between the QBO and ENSO influences to water vapor entry.

The results idea is of great interest to potential readers and worth it for publication. However, the manuscript has 3 mains issues, which are a lack of honest motivation of the study, methodological failure and finally the presentation of the result issue that I have detailed in the major and specific comments. Major revisions are needed in order to make the paper suitable for publication. There are some additional points that need to be clarified. I apologize if I misunderstood something.

**2. Major comments**
i. *Honest motivation of the study*: The concern is the main motivation of the paper. From Page 2, L13-35 and Page 3, L1-10, the discussion is not clearly and honestly reported. The authors are using the result from Garfinkel et al. 2018 based on CCMs (geosccm) to argue about questionable "nonlinear" ENSO response. The result of "nonlinear" ENSO impact on water vapor is still an open question as the observations does not show this nonlinearity. In addition, the models used in Garfinkel et al 2018, 2021 are based on spontaneous generated QBO, which does not reach the tropopause and does not have the same QBO phases and strength as the observations. This lead to wrong water vapor modulations by the interplay between the QBO and ENSO in the lower stratosphere, therefore, to questionable result of Garfinkel et al 2018, 2021. The discussion about about "El Nino and La Nina can lead to moistening" is still very questionable, knowing the inability of CCMs to reproduce a descent tape recorder (Keeble et al 2020). The look like circling discussion (e.g. dog biting its tail) about the ''nonlinearity'' of ENSO need to be discussed clearly as it still needs to be proven with the observations rather than it's already an established results. As we know CCMs have several issues when it comes to water vapor entry in the lower stratosphere.

These comments seem to reflect confusion on the part of the reviewer. We clearly show in this paper (and also in Garfinkel et al 2018, 2021) that the ENSO$^2$ regressor is important in observations. Garfinkel et al 2018, 2021 also show it is important in some models (including both GEOSCCM and WACCM) though not all models. Taking these three papers as a whole, we agree that we haven't "proven" its importance (nor do we state that we have). Rather,

we demonstrate in this paper that adding it to observations closes the gap between a simple MLR and more advanced techniques.

Second, we fully agree that CCMs have several issues when in comes to water vapor entry in the lower stratosphere, including too warm of a cold point (in most models), a lack of downward propagation of the QBO, and too weak interannual variability of cold point temperatures. These limitations are discussed both in Garfinkel et al 2021 and also in this paper.

Finally, one important remark is the Authors are using their OWN multiple regression model, which is not the same as the Dessler et al. 2014, Diallo et al, 2018 and Tian et al. 2019. There are as many as different MLR in terms of predictors, including a dynamical or fixed lag and solver. Just note that the Diallo et al. 2018 method is not a simple MLR where you can predefine a fixed lag as your regression but a multivariate hybrid method. In order word you have used your OWN regression, therefore, you should be that general as even your regression has issues.

We are also a bit confused by this comment. MLR is a standard statistical technique used in dozens of scientific fields, and taught in statistics courses. Its basic ingredients are not particularly complicated. There isn't room for subjectivity in this setup.

That being said, there are decisions that must be made as to what predictors to include (which we discuss in our paper), and also about which lags to include (which we also discuss in our paper), and these can affect the results. The precise choice of which lag to use for e.g. the QBO does indeed matter somewhat, though we are clear in our paper as to what lag we use and thus our results are fully reproducible.

ii. *Methodological failure:* The regression model used here is failing to reproduce the ENSO (El Nino and La Noina as well) induced impact on water vapor variability (Figure 6). The QBO coefficient also looks strange. The ENSO-square even looks like a second QBO coeficient. Actually, the ENSO impact on H2O structure is a horseshoe pattern as shown in Konopka et al 2016 and Avery et al. 2017.

MLR is a standard statistical technique, and all steps needed to reproduce our results are included in the paper and in the code made available. Neither the smallness of the regression coefficient for ENSO, nor the relatively larger $ENSO^2$ regression coefficient, are bugs.

Figure 11 of Konopka et al 2016 clearly shows that the zonally asymmetric structure is present at theta=390K though absent at theta=420K/450K. Our study shows a zonally asymmetric structure as well at 82hPa in SWOOSH data in the MLS period. Thus, there is no contradiction to our studies, though we have added to the discussion that Konopka et al 2016 found such a zonally asymmetric feature at 390K.

Figure 1 of Avery et al 2017 also shows a moistening but with zonally structure in December 2015 when extreme El Nino conditions were present. This again is entirely consistent with our paper.

Apparently, the regression model used here only is not capturing ENSO look-like impact on H2O entry. When multiplying QBO impact by ENSO impact (QBOxENSO), the result looks like an ENSO impact on H2O pattern. The Figure 7 and the large residual over the entire period (trained and tested) in Figure 9 both corroborate the failure of Ziv et al MLR model.

MLR is a standard statistical technique, and all steps needed to reproduce our results are included in the paper and in the code made available. The difference in skill between the MLR model with only ENSO and the QBO vs. the more advanced techniques highlights the nonlinearity whereby ENSO affects entry water.

The QBO signal from their MLR is also questioning.
Actually, the QBO signal in our MLR agrees well with previous work, and accounts for ~0.015 ppmv (m/s)$^{-1}$, or around 0.4 ppmv peak to peak (See our figure 9 and Dessler et al 2013 figure 5).

So, major analysis are need to investigate this failure before concluding.
Possible diagnostics are: First, it would be great to see how well your MLR and ML are able to capture the altitude-time cross section of the tropical H2O variability induced by the QBO and ENSO (5S-5N mean of their effect).
This paper is focused on entry water vapor, not water vapor higher up. Please see the title of our paper.

Second, estimate the R-square error of the residual.

This information is included on Figure 5.

Third, verification of the used ENSO proxy if it is not too small and also especially in the manuscript (Page 7, L2), you stated using NINO3.4 from ERSSTv5 data with a 1981-2010, while the analysis period is 1994-2019.

Results are essentially identical if we change the base period.See the Nino3.4 index below for two different base periods.

[Figure]

Regarding the ML, the different supervise learning method are barely described in the manuscript.

We improved the explanations for the different ML methods.

The other main issue is the training and testing period of the Machine learning. The authors did not use an independent training date set for testing the performance of the machine leaning model. The approach of using the same data randomly sampled and divided into 5 fold won't help to assess the performance of the ML model. This is a serious issue. You should show at least show the ML performs in the unseen test sample to disclose over-fitting issues etc.

This is simply not true. We do use an independent training set. However due to the limited record length, we elect to shuffle which part of the available record length is left out, and then train on the rest, rather than subjectively deciding which period to leave out. This is described in Section 2. We very clearly show in figure 5 how the ML performs in the unseen test sample.

The lag used in the manuscript is not clear if it is observed one or the one from the CCMs. Please clearly describe each method and explain what you have done.

We use the same lag for all CCMIs models, and a different lag for all CMIP6 models, due to the difference in level used to track entry water. This is stated in section 2.

Finally, the cold point temperatures are very well negatively correlated with the H2O as the latter is determined by freeze-drying process (Fueglistaler & Haynes, 2005; Fueglistaler et al. 2013; Poshyvailo et al 2016; Grandville & Birner al 2016). The CPT as H2O then are both modulated by the climate modes of natural variability, including QBO and ENSO. So comparing CPT and the QBO and ENSO as predictors is not making sense at all. Since early findings, we know the strict relationship between H2O and CPT. One should use the CPT if one would like to predict H2O entry but when it comes to separating and understanding different contributions to H2O inter-annual variability, it does not make sense.

This is exactly our point! We are using CPT as an upper bound on the role of large scale processes for entry water, and we then try to explain the role of the QBO and ENSO for

interannual variability of this entry water. This is explained both in section 2 and the discussion.

iii. *Presentation of the result issue*: The structure and presentation of the results have issues which need to be improved. The authors discussed about CCM2 (Page 4, L14-18), while they are not using it. I recommend to remove this part but clarify the CCM1 representation of QBO (nudged or spontaneous) and SSTs (modelled or observed), which missing here. For instance, EMAC has also the nudged QBO, which is not mentioned, but you emphasise the WACCM water vapor coefficient are due to the nudged. So it should be the same for EMAC bot no. In addition, the level of 82.54hPa used here is not a reference level, knowing that model like WACCM has a high tropopause (about 90 hPa). I would recommend to do these analysis of the manuscript at one fixed level 70 hpa for all data sets, which is actually the reference level where tropospheric influence is separated from the stratospheric ones. They could interpolate all the data at the 70hpa level.

We believe it is important to explain why we don't use the latest CCMI models, as some modeling groups have gone to great lengths to improve their model since CCMI phase 1 ended.

We are using the Ref- C2 simulations, so all SSTs are based on models.

Thank your for bringing to our attention that EMAC also nudged their QBO. We are glad to correct this! It turns out this supports our arguments, as EMAC has the third highest correlation between entry water and the QBO (after only the NCAR models), but a particularly poor regression coefficient due to a poor simulation of interannual variability in entry water. We now clarify that three models nudge their QBO in the methods section, and have updated the results section to reflect the fact that the QBO is nudged in EMAC too.

Most models archive entry water at 80hPa, now noted. This difference between 80hPa and 82hPa is likely insubstantial.

**3. Specific comments**
a) Page 2, L9, Please add citations: Punge et al. 2009, Niwano et al. 2003 & Diallo et al. 2018.

We have added citation to Niwano et al and and Diallo et al

b) Page 2, L13-20, please discuss the zonal mean struture of the ENSO induced impact on H2O based on the observation that has been found in previous litterature (Randel et al. 2009, Calvo et al 2010, konopka et al 2016). This is what so far the truth.

Calvo et al 2010 and Konopka et al 2016 were already cited. We have added Randel et al 2009

c) Page 2, L21-30 please rephrase the entire paragraph. The claimed ``nonlinear ENSO impact on H2O'' still need to be proved in the observations, therefore, it should not be presented as ground true the same models are pointed out having issues with the QBO, which stuck at 50hPa, not realistic QBO phases compare to observed one. Conclusions from these that struggle to reproduce the tape record should be take with caution, which is not the case here.

As discussed above, we do not imply nor claim the nonlinear impact of ENSO on entry water has been proved. We also already acknowledged earlier on page 2 the issues models have with entry water.

d) Page 2, L34-35 please remove the citations "Diallo et al. 218; and Tian et al. 2019" as they are not simpl MLR as you frame here.

removed

e) Page 3, L3, this statement "First, Garfinkel et al 2018 found ...ENSO is nonlinear" needs to rephrase and made clear by precising it is model based and not consistence with the observations finding yet.

Garfinkel et al 2018 provided observational evidence too (as did Garfinkel et al 2021).

f) Page 3, L1-10, please discuss also these papers: Evans et al 2014; Brinkop et al 2016, Less et al 2012, Diallo et al. 2018 about the interplay between the ENSO and QBO impact on H2O entry.

We have added Diallo et al 2018 and Brinkop et al 2016. We were unable to find the relevant Evans et al and Less et al papers.

g) Page 4, L1-10 please precise that you are using the CCMI phase 1 models. In addition, please explain the model issues about getting the QBO right in the CCMI-1 and CMIP6 models.

added

h) Page 4, L14-27 please remove the CCM-2 discussion. It is confusing the reader as any way you are focussing on CCMI-1. Please emphasize the models ability in reprodcuing ENSO and QBO impact on the tape recoder and the uncertainty that induces in the H2O entry.

see response to the general comment about the CCMI phase 2 models. We have added a sentence about the QBO.

i) Page 5, L1: Please do the analysis at 70hPha for all the plots.
see response to the general comment

j) Page 5, table 2, please the QBO and SST infor mation for each model in the table.
QBO information added to the caption.

k) Page 6, the captions of Figure 1 are not very clear. Please clarify them.

clarified

l) Page 6, the Figure 1 should be done at 70hPa for all models and observation.

see response to the general comment

m) Page 7, L1-2, please clarify "...ERSSTv5 data with a 1981-2010 base period".

Results are essentially identical if we change the base period.See the Nino3.4 index above for two different base periods.

n) Page 8, L5-15, a clear description of the different supervised learning regression are need here to improve clarity of the method.

We improved the explanations for the different ML methods.

o) Page 9, L3-4, please remove the citations Dessler et al 2014 and Diallo et al. 2018 as you are not using their models or out put of their models for comparison. In addition, your regression model has issues in reproducing the ENSO and potentially QBO impact structure on H2O (Figure 6); tape recorder plot of QBO and ENSO induced impact on H2O and has large residual too.

see response to the general comments.

p) Page 10, L1-8, the approach used here to test the performance of the model is an issue as it you're not test the ML on unseen data for test set. How the overfitting or under fitting issues are evaluated then? It would be great to add a figure in the main paper or supplement about the ML performance showing trained period and unseen predicted H2O period. Please clarify also the training period.

see response to the general comments

q) Page 11, L13-24, Here the authors should not generalise about the MLR and its results but precise it is THEIR MLR with its limitations. The whole paragraph nee to be revise after evaluating the ability of their MLR to capture QBO and ENSO induced impact on H2O as altitude-time tropical cross-section.

see response to the general comments

r) Page 11, L25-34, the SHAP method comes out off blue. Please clarify and rephrase the paragraph

We include a sentence introducing the technique

s) Page 13, the coefficient of their MLR in Figure 6a & b are wrong as well as the Figure 7. ENSO impact on H2O is not similar to classical method results. Please evaluate clearly, why? In addition, the Figure 6 d e.g. ENSO squarred is very likely a QBO signal as you are not using two QBO index with a chosen lag for all latitude bin this may impact you MLR results. The MLR needs to be evaluated before drawing any useful conclusion here.

see response to the general comments

t) Page 13, L1-8, QBO being predominate in modulating H2O entry have been already found by Diallo et al. 2018 and confirmed by Tian et al 2019. Please discuss them.

We have added that our results are consistent with Tian et al and Diallo et al

u) Page 14, L2-5, knowing the model inability of reproducing the QBO down to the low stratosphere, it is a bit strange that the author aims at evaluating the model ability to capture the interplay between QBO and ENSO impact on H2O entry. Please rephrase the entences.

The reviewer appears to be confused, as we don't consider the ENSO impact in this section. This is stated clearly.

v) Page 17, L16, the zonal structure temperature and H2O anomalies find in previous studies (Randel et al 2009, konopka et al 2016) is a result of the averaged between a region of updraft (cold) and subsidence (warm).

We have added reference to Konopka et al.